# Nucleic Acid Sensing in the Tumor Vasculature

**DOI:** 10.3390/cancers13174452

**Published:** 2021-09-03

**Authors:** Adrian M. Baris, Eugenia Fraile-Bethencourt, Sudarshan Anand

**Affiliations:** 1Department of Cell, Developmental and Cancer Biology, Oregon Health & Science University, Portland, OR 97239, USA; baris@ohsu.edu (A.M.B.); frailebe@ohsu.edu (E.F.-B.); 2Department of Radiation Medicine, Oregon Health & Science University, Portland, OR 97239, USA; 3Knight Cancer Institute, Oregon Health & Science University, Portland, OR 97201, USA

**Keywords:** nucleic acid sensors, TREX1, cGAS, STING, RIG-I, tumor angiogenesis, vascular normalization, vascular inflammation, endothelial cells, tumor microenvironment

## Abstract

**Simple Summary:**

Our cells can recognize DNA or RNA from pathogens, such as viruses. The proteins that recognize these nucleic acids are known as nucleic acid sensors. Upon activation, they trigger immune responses that result in the elimination of the infected cells. Recent research has shown how we can mimic this process in cancer and recruit immune cells against the tumor. Among the different cell types within a tumor, endothelial cells that line the blood vessels play a main role as conduits for nutrients and oxygen and highways for the immune cells. In this review, we discuss two different nucleic acid sensors—the three-prime repair exonuclease 1 (TREX1) and the retinoic acid-inducible gene 1 (RIG-I)—and how they play a role in endothelial cells. We present some approaches to target these pathways within the cancer blood vessels to disrupt the blood supply and attract immune responses to cancers.

**Abstract:**

Endothelial cells form a powerful interface between tissues and immune cells. In fact, one of the underappreciated roles of endothelial cells is to orchestrate immune attention to specific sites. Tumor endothelial cells have a unique ability to dampen immune responses and thereby maintain an immunosuppressive microenvironment. Recent approaches to trigger immune responses in cancers have focused on activating nucleic acid sensors, such as cGAS-STING, in combination with immunotherapies. In this review, we present a case for targeting nucleic acid-sensing pathways within the tumor vasculature to invigorate tumor-immune responses. We introduce two specific nucleic acid sensors—the DNA sensor TREX1 and the RNA sensor RIG-I—and discuss their functional roles in the vasculature. Finally, we present perspectives on how these nucleic acid sensors in the tumor endothelium can be targeted in an antiangiogenic and immune activation context. We believe understanding the role of nucleic acid-sensing in the tumor vasculature can enhance our ability to design more effective therapies targeting the tumor microenvironment by co-opting both vascular and immune cell types.

## 1. What Are Nucleic Acid Sensors?

In order to protect themselves from outside pathogens and other agents, organisms have developed two interlinked forms of defense systems: an innate and an acquired immunity. The innate immune response reacts rapidly to an infection, which can often exponentially multiply long before the adaptive immune response is able to take effect [1]. These responses should be tightly regulated to prevent dysfunction or damage to the host. The innate immune system responds to damage-associated molecular patterns (DAMPs), which are recognized by pattern recognition receptors (PRRs). The activation of PRRs typically induces a downstream type I interferon (IFN-I) and cytokine response. Nucleic acid sensors are a specific type of PRR that recognize pathogen-derived cytosolic nucleic acids and activate downstream signaling cascades, which produce a pro-inflammatory response. This response is critical to the ability of nucleic acid sensors to stop pathogens in their tracks [2]. Nucleic acid sensors are unique in that they can recognize and differentiate exogenous and endogenous nucleic acids [2,3]. A number of nucleic acid sensors (NASs) play a prominent role in inflammation [4]. Recent studies have shown that NASs play a large role in endothelial function and dysfunction [5]. The endothelium regulates vascular tone and growth [6] and is critical in pathologies, such as viral infections, cardiovascular disease, and cancer. A better understanding of the role NASs play in endothelial function is key to improving our knowledge of tumor angiogenesis, as well as cardiovascular and infectious diseases. Several outstanding reviews [3,7,8,9,10] discuss the sensors outlined in Table 1. In this review, we will focus on the DNA sensor three-prime repair exonuclease 1 (TREX1) and RNA sensors in the retinoic acid-inducible gene 1 (RIG-I)-like receptors (RLRs) pathway (summarized in Figure 1).

## 2. What Is TREX1?

DNAse III, or TREX1, is an exonuclease that degrades exogenous DNA. TREX1 is a member of the DnaQ family of 3′→ 5′ exonucleases. These exonucleases are known for three conserved sequence motifs: Exo I, II, and III, which are essential for exonuclease function. The TREX1 protein contains a C-terminal domain of about 75 amino acids and a non-repetitive, proline-rich region that is not seen in the other TREX protein, TREX2 [11,12]. TREX1 degrades single-strand DNA (ssDNA), double-strand DNA (dsDNA), and single-strand RNA (ssRNA) in the cytosol [13,14]. Cytosolic DNA activates the cGAS–STING–TBK1 pathway and leads to the downstream activation of IRF3 and IRF7, which induce type I interferons [15]. By degrading cytosolic DNA, TREX1 removes the substrates for cGAS, thus dampening the nucleic acid sensor response. Therefore, TREX1 is thought to be a negative regulator of interferon signaling, and broadly, autoimmune diseases. TREX1 interacts with poly (ADP-ribose) polymerase (PARP) and is a facilitator of its nuclear translocation and activity [16], suggesting a putative role in DNA damage responses. Indeed, other studies have shown DNA damage can increase TREX1 expression in a dose-dependent manner [17,18]. In addition to nucleic acid-sensing, TREX1 is also involved in the regulation of an ER-resident enzyme, oligosaccharyltransferase (OST), which contributes to its role in immune regulation and inflammation [19,20].

## 3. What Are the Biological Roles of TREX1?

### 3.1. DNAse Dependent Functions in the cGAS-STING Axis

As outlined above, TREX1 prevents the activation of the cGAS-STING pathway. cGAS is a cytosolic cyclic gAMP synthase that is typically in a catalytically inactive, autoinhibited state. However, in the presence of cytosolic DNA, cGAS binds to DNA and undergoes a conformational change which catalyzes the synthesis of cyclic GMP-AMP (cGAMP). cGAMP binds to STING, an endoplasmic-reticulum resident membrane adaptor. This induces a conformational change that activates STING and causes it to traffic to the Golgi, during which TBK1 is activated. The phosphorylation of TBK1 activates its kinase activity, leading to phosphorylation of IRF3, and induces the expression of interferons and inflammatory cytokines [21]. In 2010, it was discovered that TREX1 suppresses the interferon response triggered by HIV. In TREX1-deficient mouse cells as well as human immune cells with TREX1 RNAi knockdown, HIV infection was able to produce a type I IFN response that inhibited HIV replication and spread [22].

Interestingly, mice deficient in DNAse II, a similar enzyme, die during embryonic development through inflammatory disease. In the absence of DNAse II, apoptotic cells are engulfed normally during development but are not digested in the lysosomes of macrophages [23]. Therefore, apoptotic self-DNA accumulates in these macrophages and triggers type I interferon responses. This phenotype is rescued by the loss of STING, as cytosolic DNA was unable to trigger inflammatory cytokine production in the STING^−/−^ DNAse II^−/−^ double knockout mice [23]. Outside of development, other pathophysiological stimuli can also lead to aberrant activation of cGAS-STING signaling responses. For instance, radiation-induced cytosolic DNA is a potent trigger of type I interferon responses in the tumor microenvironment [24]. There is evidence suggesting that the activation of STING can also contribute to radiation-induced toxicity, such as cardiac inflammation and fibrosis [25,26,27]. There are also other pathways that activate cGAS-STING signaling. Recently, Coquel et al. [28] discovered that the dNTPase SAMHD1 promotes the degradation of nascent DNA at stalled replication forks. This function appears to co-opt the exonuclease activity of MRE11, thereby activating the ATR–CHK1 checkpoint and enabling the cells to recover and restart the stalled replication forks. The depletion of SAMHD1 led to an increase in the accumulation of ssDNA fragments in the cytosol and the subsequent activation of the cGAS–STING pathway. These findings suggest that TREX1 and other enzymes with nuclease activity play an important role in the context of the cGAS-STING signaling pathway’s regulation of inflammation in tissues.

Interestingly, mice deficient in DNAse II, a similar enzyme, die during embryonic development through inflammatory disease. In the absence of DNAse II, apoptotic cells are engulfed normally during development but are not digested in the lysosomes of macrophages [23]. Therefore, apoptotic self-DNA accumulates in these macrophages and triggers type I interferon responses. This phenotype is rescued by the loss of STING, as cytosolic DNA was unable to trigger inflammatory cytokine production in the STING^−/−^DNAse II^−/−^ double knockout mice [23]. Outside of development, other pathophysiological stimuli can also lead to aberrant activation of cGAS-STING signaling responses. For instance, radiation-induced cytosolic DNA is a potent trigger of type I interferon responses in the tumor microenvironment [24]. There is evidence suggesting that the activation of STING can also contribute to radiation-induced toxicity, such as cardiac inflammation and fibrosis [25,26,27]. There are also other pathways that activate cGAS-STING signaling. Recently, Coquel et al. [28] discovered that the dNTPase SAMHD1 promotes the degradation of nascent DNA at stalled replication forks. This function appears to co-opt the exonuclease activity of MRE11, thereby activating the ATR–CHK1 checkpoint and enabling the cells to recover and restart the stalled replication forks. The depletion of SAMHD1 led to an increase in the accumulation of ssDNA fragments in the cytosol and the subsequent activation of the cGAS–STING pathway. These findings suggest that TREX1 and other enzymes with nuclease activity play an important role in the context of the cGAS-STING signaling pathway’s regulation of inflammation in tissues.

### 3.2. DNAse Independent Functions

TREX1 has a DNAse independent function that suppresses immune activation through the regulation of oligosaccharyltransferase (OST) activity. TREX1 is phosphorylated during mitosis, which disrupts its interactions with the OST complex without affecting its DNAse activity [20]. 

### 3.3. Emerging Functions of TREX1

TREX1 has a wide range of functions in immunity, DNA damage, and cancer. While it has been widely studied, there are still many new functions still being discovered. For example, TREX1 has been shown to inhibit cGAS activation at micronuclei through the degradation of micronuclear DNA [29]. In addition, another recent study showed that cGAS-DNA phase separation sequesters cGAS and prevents access to TREX1. This enhances cGAS DNA sensing activity [30]. TREX1-deficient mice exhibit reduced survival rates and develop inflammatory myocarditis [31]. This often leads to cardiomyopathy and circulatory failure. These mice also develop lethal interferon-driven autoimmune disease; however, the cGAS deficiency is protective in this instance [32].

### 3.4. TREX1 Associated Human Disease

Mutations in TREX1 have been linked to several disorders. For example, TREX1 mutations are found in patients with Aicardi–Goutières syndrome (AGS), which is an encephalopathy that results in severe neurological dysfunction [33,34,35]. Patients with AGS associated with a TREX1 mutation typically experience neurological defects, such as dystonia, seizures, cortical blindness, and progressive microcephaly, and are more likely to be affected at birth. There are 5 unique TREX1 mutations that contribute to AGS [36]. These include a G314A transition that results in a nonconservative R to H substitution predicted to be involved in protein dimerization, a missense T602A mutation in the ExoIII motif, and two protein-truncating mutations. Type I interferon signaling is shown to be upregulated in most TREX1 AGS patients. TREX1 mutations and increased type I interferon signaling have also been associated with familial chilblain lupus (FCL) [37,38,39].

## 4. How Does TREX1 Function Impact the Vasculature?

Hereditary vascular retinopathy (HVR) is another rare disorder linked to mutations in TREX1 and eventually leads to blindness [40]. It is a microvascular endotheliopathy without any obvious immunological symptoms; however, vascular integrity is compromised. TREX1 mutations are also associated with another syndrome, retinal vasculopathy with combined leukodystrophy (RVCL), where capillaries in the retina degenerate leading to vision loss and brain pathology that shows DNA damage [35]. We observed similar functional consequences in murine models both in a neonatal ocular angiogenesis model and in the tumor vasculature with miR targeting of TREX1. Our lab has demonstrated that TREX1 silencing exacerbates DNA damage and cell death, inhibits angiogenesis, and induces inflammatory cytokines [17]. Complementary to our observations in the vasculature, Vanpouille-Box et al. showed that TREX1 in tumor cells is induced by high-dose radiation and the inhibition of TREX1 synergizes with the immune checkpoint blockade [18,41].

Going beyond TREX1, there is emerging evidence for the function of downstream pathways in the vasculature. STING is expressed in both normal and tumor vasculature [25,42,43,44]. STING signaling in the high endothelial venule has been shown to contribute to an endothelial–lymphocyte interaction [45]. Endothelial cell-derived type I IFNs initiate antitumor responses before dendritic cells and CD8^+^ T-cells infiltrate the TME and can determine the magnitude of overall immunity [42]. In humans, mutations in the STING-encoding gene, *TMEM173*, results in a fatal vasculitis, termed STING-associated vasculopathy with onset in infancy (SAVI) [46,47]. Signaling mediators downstream of TREX1, TBK1, and IRF3 were found to be necessary proangiogenic factors in a high-throughput genomic screen of endothelial cell activity [48]. Mice deficient in TBK1 exhibit immune cell infiltrates and an increase in susceptibility to LPS-induced lethality as well as a decrease in IFN-β and T-cell expression [49]. IRF3-deficient mice experience an altered IFN response during influenza infection [50]. They also experience an influx of granulocytes in the lung and a decrease in the activation of the adaptive immune response. While these lines of evidence argue for a type I interferon-dependent process in driving inflammation, more recent work has also established a direct correlation between STING and T-cell trans-endothelial migration. STING^−/−^ vasculature produced decreased CXCL10 in response to TNFα, resulting in reduced T-cell migration across the endothelium [51].

In summary, it is clear that the regulation of type I interferons is an intricate process with several checks and balances starting with the STING pathway and many upstream regulators, such as cGAS, TREX1, SAMHD1, etc. Complex biological mechanisms contribute to the accumulation of cytosolic DNA fragments during development and disease that can eventually trigger these nucleic acid sensors. The pathological consequences of activating these DNA sensors are also well recognized in diseases such as SLE. Understanding these pathways will not only provide biological insight into how these pathologies develop but will also provide actionable drug discovery pathways, such as STING antagonists, that can be effective for human diseases.

## 5. What Is the RLR Family of Intracellular RNA Sensors?

In addition to DNA, PRRs can also recognize RNA in the cytoplasm. The RIG-I-like receptor family (RLR) contains three RNA sensors: retinoic acid-inducible gene-I (RIG-I), melanoma differentiation-associated protein 5 (MDA5), and laboratory of genetics and physiology gene 2 (LGP2). These three proteins share a C-terminal domain (CTD) and a helicase domain (HD), but only RIG-I and MDA5 bear the effector domain known as the caspase activating and recruiting domain (CARD) in the N-terminal. Therefore, only RIG-I and MDA5 trigger the IFN-I response through interactions with mitochondrial antiviral signaling proteins (MAVSs) [52,53,54].

## 6. What Are the Substrates (Foreign and Self) for RLRs?

RLRs are activated by cytosolic RNA. RIG-I and MDA5 are similar in structure and function, but they recognize different RNA structures. RIG-I is preferentially activated by blunt-ended 5′ppp short RNAs, which bind to the CTD [55,56]. The CTD of RIG-I has a pocket that specifically binds to either a 5′-PPP or a 5′-PP. In normal conditions, the CARD is bound to the HD in a repressing form. Upon RNA recognition, the base-paired region of RNA complexes with the HD of RIG-I, releasing the CARD. Thus, the stable RNA–RIG-I interaction displaces CARDs, which causes multiple RIG-I proteins to oligomerize and become accessible for MAVS signaling. One main player in this process is the E3 ubiquitin ligase TRIM25. TRIM25 ubiquitination is critical for the release of RIG-I from autorepression. To interact with MAVS in the mitochondria, the RIG-I complex, RIG-I/14-3-3ε/TRIM25, mediates the redistribution or “translocation” of RIG-I from the cytosol to the intracellular membrane compartments. There, RIG-I binds to MAVS through homotypic CARD–CARD interactions [57]. Once activated, MAVS recruits the tumor necrosis factor receptor-associated factors (TRAFs), which are essential for the activation of interferon regulatory factors 3 and 7 (IRF3, IRF7) and NF-kB-mediated responses [58]. Finally, the activation of RIG-I results in the expression of cytokines and IFN-I genes, which recruits innate and eventually adaptive immune cells [57].

RLRs are able to recognize self-derived RNAs, leading to either enhanced or depleted IFN responses in a context-dependent manner [56]. A recent study showed that mitochondrial DNA double-stranded breaks release mitochondrial RNA into the cytoplasm, triggering the RLR-dependent immune response. Moreover, following cellular irradiation, mitochondrial DNA breaks synergize with nuclear DNA to promote the immune response [59]. These emerging studies highlight the potential of RIG-I activation without extrinsic pathogens and could potentially explain the ‘sterile inflammation’ in tissues.

## 7. What Are the Biological Roles of the RLR Family RNA Sensors?

While RIG-I and MDA5 drive very similar signaling pathways, they do differentially induce type I IFN responses to different pathogens [53]. For example, while RIG-I is activated most potently in response to negative-strand viruses, such as the influenza virus [60], MDA5 is activated in response to positive-strand viruses, such as the hepatitis D virus [61]. In addition, animal models show that RIG-I and MDA5 have functional differences in vivo, as well as distinct molecular immune functions [52].

### 7.1. Phenotypes in Knockout Mice

RIG-I knockout mice show a colitis-like phenotype, reduced Peyer’s patches, increased effector T-cells, and decreased naïve T-cells [62]. MAVS and MDA5 knockout mice lose type I interferon production and suffer early mortality in response to infection with the Coxsackie B virus (CVB), which has been associated with myocarditis [63]. In a study of RLRs in the West Nile virus, RIG-I x MDA5 double knockout mice lacked the innate immune response against the viral infection. Surprisingly, they did not suffer severe pathological damage in their tissues during infection, which was similar to what was found in animals lacking MAVS [64].

### 7.2. Associations with Human Disease

Singleton–Merten syndrome (SMS) is an autosomal-dominant disorder characterized by aortic calcification, skeletal abnormalities, psoriasis, as well as other conditions. Jang et al. performed exome sequencing and found gain-of-function mutations in DDX58, the gene which encodes the RIG-I protein and leads to the variable manifestation of SMS, often without the typical dental anomalies [65]. In addition, gain-of-function mutations of MDA5 have been found in SMS patients with upregulated interferon signature genes. The sustained signaling of MDA5 and RIG-I in SMS patients is possibly due to an increase in protein levels, the recognition of self-RNA, or both. It is believed that an excess of IFN-I and other inflammatory cytokines in the endothelial cells in aortic and mitral valves is critical for SMS development [66].

### 7.3. Checks and Balances on RIG-I Signaling

Several feedback inhibitory pathways have evolved to regulate RIG-I signaling in cells. For instance, the lncRNAs ATV and Lsm3b have been shown to directly inhibit the RIG-I CTD [67,68]. Other ncRNAs, such as miR-526a, indirectly impact RIG-I by downregulating CYLD, an enzyme that inhibits ubiquitination of the RIG-I CARD domain [69]. There are other miRNAs, such as miR-485, that directly inhibit RIG-I transcripts [70]. In addition, several post-translational modifications regulate RIG-I function. For example, TRIM38-mediated sumoylation, either in the CARD domain or in the CTD, can activate RIG-I [71]. Conversely, acetylation of RIG-I in the CTD at K909 is thought to prevent RIG from binding to viral RNA. Hence, CTD deacetylation by HDACs, especially HDAC6, can enhance RIG-I activation and signaling [72].

## 8. How Does RLR Function Impact the Vasculature?

It has been shown that endothelial RIG-I activation leads to endothelial dysfunction. In wild-type mice, the activation of RIG-I leads to endothelial stress, damage, and vessel impairment. After injection with a RIG-I agonist (dsRNA with a triphosphate at the 5′ end), mice experienced vascular oxidative stress and increased circulating endothelial microparticle (EMP) numbers, indicating endothelial dysfunction. In addition, after stimulation with a RIG-I agonist, both human coronary endothelial cells (HCAECs) and endothelial progenitor cells (EPCs) showed increased reactive oxygen species formation, and HCAES increased the production of proinflammatory cytokines [73]. Similarly, the stimulation of MDA5 led to endothelial apoptosis, the formation of reactive oxygen species, and the release of pro-inflammatory cytokines. MDA5 activation in mice, similar to RIG-I, leads to vascular oxidative stress and an increase in circulating endothelial microparticles and endothelial progenitor cells. In addition, chronic MDA5 stimulation exacerbated atherosclerosis [74]. Similar to these pathways, the activation of another RNA sensor, toll-like receptor 7, also leads to vascular inflammation and impaired vascular growth [75]. 

## 9. How Can Viruses Trigger Vascular Dysfunction through the RIG-I Pathway?

Vascular endothelial cells line the inner surface of blood vessels and provide a barrier between organ systems and blood vessels. This makes them critical during viral infections. The viral infection of endothelial cells allows the virus to disperse to other organs, and provides a reservoir for long-term persistence. In addition, viral replication and the immune response in the endothelium lead to increased tissue permeability as well as inflammation. Altogether, these changes drive vascular and pulmonary disease that further exacerbates the viral disease [76]. Endothelial activation and dysfunction have been shown to serve a necessary mechanistic role in the pathology of severe influenza [77,78]. For example, RLRs and TLR7 signaling have been shown to be necessary for cell survival and for restricting virus growth in mice [79]. RLRs, as well as other aspects of the innate immune system, were found to induce antiviral innate and adaptive immune responses. Interestingly, in some cases, the influenza virus has been shown to co-opt the signaling of TLR7 and RIG-I [80].

Other viruses rely on endothelial cells for their replication and host response. Particularly relevant of late, endothelial cells have been hypothesized to be essential mediators of pathology in SARS-CoV-2 infections [81]. After the initial phase of infection, some patients experience an overactive inflammatory response which leads to lung damage and increased disease severity. It was proposed that SARS-CoV-2 may cause pulmonary vascular changes based on clinical observations. Thus, the endothelial cell injury and dysfunction caused by SARS-CoV-2 likely contributes to the life-threatening complications of COVID-19 [82]. While the mechanistic role of nucleic acid sensors in the vascular pathologies induced by SARS-CoV-2 remains to be elucidated, they have been well characterized in the vascular inflammatory response to other infectious pathogens.

## 10. How Can We Target Nucleic Acid Sensors to Diminish Tumor Angiogenesis?

Tumor angiogenesis involves a number of pro-angiogenic factors in the tumor microenvironment driving the formation of new blood vessels [83]. The tumor vasculature is characterized by disorganized and immature vessels that permit the extravasation of tumor cells leading to distant metastasis. Decades ago, it was found that tumor angiogenesis is correlated with more metastatic disease in breast cancer [84]. In this process, ECs were long thought to only provide support for tumors, without playing an active role. However, we now appreciate that endothelial cells do regulate a wide variety of cancer cell functions. Secretions from quiescent ECs are able to reduce cancer cell proliferation and invasiveness. In fact, altering the secretome of ECs inhibits their ability to suppress cancer progression [85]. In addition to paracrine effects on tumor cells, the secretome of ECs also exerts autocrine effects. For instance, IFN-β is an antiangiogenic cytokine that targets endothelial cells [86]. Endothelial cells are also a major source of type I IFNs, highlighting a potential feedback inhibitory loop where ECs secrete type I IFNs that inhibit angiogenesis [87,88,89].

The notion of anti-angiogenic cancer therapy relies on the idea that removing tumor vasculature will prevent nutrients from entering the tumor, thus leading to an effective therapy. Conventional anti-angiogenic drugs often target agents that promote blood vessel formation and are overexpressed in tumors, such as VEGF-A. However, anti-angiogenesis treatment cannot eradicate the tumor on its own. The use of chemotherapy or immunotherapy in conjunction with anti-angiogenic treatment can provide a more effective strategy [90,91]. Several studies have shown that combining anti-angiogenic therapy with conventional chemotherapy can lead to improved clinical outcomes [92,93,94,95,96]. Interestingly, among the nucleic acid sensor pathways, STING activation appears to have this dual function—serving to normalize tumor vasculature as well as improve the immune microenvironment. STING activation increased pericyte coverage and increased effector T-cell migration across the endothelial barrier, thereby enhancing antitumor immunity. This approach synergized well with a VEGF inhibitor, and displayed the utility of combining anti-angiogenic drugs with vascular immune therapy [97].

We showed that the inhibition of endothelial TREX1 through miR-103 and siRNA decreased angiogenesis and tumor growth [17]. Similarly, our recent efforts in the lab have focused on the role of RIG-I in tumor vasculature. The use of a 5′PPP containing siVEGF RNA was shown to provide both anti-angiogenic therapy and activate RIG-I, leading to an antitumor effect in a murine model of lung cancer [98]. IRF-1, a known tumor suppressor, has also been shown to inhibit angiogenesis. This has been attributed to a splicing variant involving exon 7 and has implications for anti-angiogenic cancer therapies [99]. Even recently, Myct1 was identified as a transcription factor in the tumor endothelium and was shown to play a dual role in tumor angiogenesis and tumor immunity [100]. In summary, these studies provide an overview of the emerging rationale and approaches for targeting RNA sensors to decrease angiogenesis, enhance vascular normalization, and improve the antitumor immune microenvironment.

## 11. Conclusions and Future Perspectives

Here, we have highlighted how two important nucleic acid sensors—the DNA sensor TREX1 and the RNA sensor RIG-I—can drive endothelial dysfunction and inflammation. Targeting these pathways not only impacts angiogenesis but also elicits potent immune responses. Therefore, these are attractive targets for drug discovery across therapeutic areas ranging from cardiovascular disease to infectious diseases, as well as cancer. In cancer anti-angiogenic treatments, targeting nucleic acid sensors may also stimulate an immune response. Returning the tumor vasculature to a more normalized state is thought to reverse the immunosuppressive tumor microenvironment and allow the tumor to be treated more effectively with chemotherapy, immunotherapy, and radiation. This has been observed with VEGF inhibitors in tumors; however, the success is limited due to immune evasion and the development of therapeutic resistance. Anti-angiogenic drugs can be combined with immunotherapies to produce a more potent antitumor response, and have been shown to have clinical potential. We anticipate that with more insight into these cytosolic nucleic acid-sensing pathways in the tumor vasculature, we will be able to design versatile agents that can both inhibit angiogenesis as well as actively stimulate innate and adaptive immune responses.

## Figures and Tables

**Figure 1 cancers-13-04452-f001:**
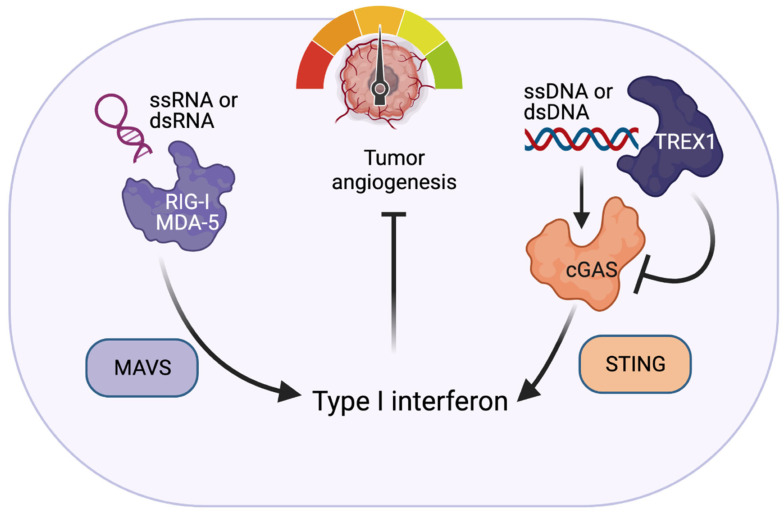
Nucleic acid sensors in endothelial cells. RNA sensors RIG-I/MDA5 respond to exogenous RNA and lead to a type I interferon response via the MAVS pathway in ECs. Conversely, TREX1 degrades ssDNA or dsDNA in the cytosol, thereby preventing activation of the cGAS-STING pathway that activates type I interferons. Activation of the RIG-I pathway or inhibition of the TREX1 pathway will inhibit tumor angiogenesis by upregulating interferon responses.

**Table 1 cancers-13-04452-t001:** Summary of nucleic acid sensor substrates and functions.

Gene	Protein	Substrate	Function	Reference
Cgas	cGAS	dsDNA	Apoptosis	[10]
TLR9	TLR9	RNA-DNA hybrid	Inflammation	[10]
TREX1	TREX1	dsDNA	Immune suppression	[8]
AIM2	AIM2	dsDNA	Pyroptosis	[10]
IFI16	IFI16	dsDNA	Pyroptosis	[8]
IFIH1	MDA5	dsRNA	Apoptosis	[8]
DDX58	RIG-I	dsRNA	Inflammation	[8]
TLR3	TLR3	dsRNA	Nectroptosis	[8]
ZBP1	ZBP1	B-DNA, Z-DNA	Necroptosis	[9]

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
