# Peer review of "Nucleic Acid Sensing in the Tumor Vasculature"

_cancers, 2021, doi:10.3390/cancers13174452_

Round 1

Reviewer 1 Report

In their review Baris et al. discussed two different nucleic acid sensors (RIGI and TREX1) and their role in endothelial cancer cells. This review is well written and easy to follow. This is clearly of interest since Type I IFN pathways are a direct focus of clinical therapy, and a better understanding of nucleic acid sensors activation, regulation and outcome will help design the best therapeutical option for patients.

However, important limitation are needed to get a comprehensive review of the subject.

The authors may write few words about other exonucleases (eg MRE11, see for instance Coquel, F., Silva, MJ., Técher, H. et al. SAMHD1 acts at stalled replication forks to prevent interferon induction. Nature 557, 57–61 (2018). https://doi.org/10.1038/s41586-018-0050-1) but also refer the reader to reviews on cGAS-STING (see for instance Constanzo J, Faget J, Ursino C, Badie C, Pouget JP. Radiation-Induced Immunity and Toxicities: The Versatility of the cGAS-STING Pathway. Front Immunol. 2021;12:680503. Published 2021 May 17. doi:10.3389/fimmu.2021.680503), which is linked with TREX1.

It would be of great interest to better develop the section “DNAse dependent function” (TREX1) page 3. Especially the last paragraph (ref 22).

The authors should merge “TREX1 knockout mouse” with the previous paragraph since it has been already mentioned.

In the section “How does TREX1 function impact the vasculature?”, based on the last sentence, I suggest the authors to develop the impact of the mentioned examples on the vasculature. Also, in my opinion, it would be of great interest to give a take home message before jumping into RLR and RIGI section.

Overall, it would be improved whether the authors give their insights, future directions and take home messages all along the manuscript.

Author Response

Author responses are in blue below the referee comments:

In their review Baris et al. discussed two different nucleic acid sensors (RIGI and TREX1) and their role in endothelial cancer cells. This review is well written and easy to follow. This is clearly of interest since Type I IFN pathways are a direct focus of clinical therapy, and a better understanding of nucleic acid sensors activation, regulation and outcome will help design the best therapeutical option for patients.

However, important limitation are needed to get a comprehensive review of the subject.

The authors may write few words about other exonucleases (eg MRE11, see for instance Coquel, F., Silva, MJ., Técher, H. et al. SAMHD1 acts at stalled replication forks to prevent interferon induction. Nature 557, 57–61 (2018). https://doi.org/10.1038/s41586-018-0050-1) but also refer the reader to reviews on cGAS-STING (see for instance Constanzo J, Faget J, Ursino C, Badie C, Pouget JP. Radiation-Induced Immunity and Toxicities: The Versatility of the cGAS-STING Pathway. Front Immunol. 2021;12:680503. Published 2021 May 17. doi:10.3389/fimmu.2021.680503), which is linked with TREX1.

We appreciate the reviewers note on the important SAMHD1 work and the STING review. We have now added this information and references (page 6).

It would be of great interest to better develop the section “DNAse dependent function” (TREX1) page 3. Especially the last paragraph (ref 22).

We have elaborated on the points to expand the scope of the section. Specifically we have included information about the cGAS/STING and other exonulceases.

The authors should merge “TREX1 knockout mouse” with the previous paragraph since it has been already mentioned.

We have merged this with the previous paragraph.

In the section “How does TREX1 function impact the vasculature?”, based on the last sentence, I suggest the authors to develop the impact of the mentioned examples on the vasculature. Also, in my opinion, it would be of great interest to give a take home message before jumping into RLR and RIGI section.

We have added a few more points and added a summary paragraph to this section.

Overall, it would be improved whether the authors give their insights, future directions and take home messages all along the manuscript.

We have added the summary for the first section providing some more insight into the cGAS/STING pathway and also edited our final section to add some conclusions before the future perspectives.

Reviewer 2 Report

This is clear-curt, comprehensive review, in which authors discuss te role of  nucleic acid sensors (NAS) within tumor endothelial cells (EC). They limit, unfortunately to 2 NAS, namely to DNA sensor, Three Prime Repair Exonuclease 1 (TREX1) and to RNA one, the Retinoic Acid-Inducible Gene 1(RIG-1). Another DNA NAS, cGAS-STING is mentioned. but not described in detail. In the table provided, 9 NAS, their substrate and function are mentio- ned only. Other data, such as the impact of NAS mutations, links with human pathology and the role in  tumor EC are of outstanding interest.

Author Response

This is clear-curt, comprehensive review, in which authors discuss te role of  nucleic acid sensors (NAS) within tumor endothelial cells (EC). They limit, unfortunately to 2 NAS, namely to DNA sensor, Three Prime Repair Exonuclease 1 (TREX1) and to RNA one, the Retinoic Acid-Inducible Gene 1(RIG-1). Another DNA NAS, cGAS-STING is mentioned. but not described in detail. In the table provided, 9 NAS, their substrate and function are mentio- ned only. Other data, such as the impact of NAS mutations, links with human pathology and the role in  tumor EC are of outstanding interest.

We appreciate the reviewer's input. We have now added some more information on cGAS/STING pathway in Page 6 and 9. We also discuss a couple of other exonuclease briefly in the context of STING activation.